# The Role of Organic Cation Transporters in the Pharmacokinetics, Pharmacodynamics and Drug–Drug Interactions of Tyrosine Kinase Inhibitors

**DOI:** 10.3390/ijms24032101

**Published:** 2023-01-20

**Authors:** Fangrui Xiu, Magdalena Rausch, Zhibo Gai, Shanshan Su, Shijun Wang, Michele Visentin

**Affiliations:** 1College of Traditional Chinese Medicine, Shandong University of Traditional Chinese Medicine, Jinan 250355, China; 2Department of Nephrology, Affiliated Hospital of Shandong University of Traditional Chinese Medicine, Jinan 250014, China; 3Department of Clinical Pharmacology and Toxicology, University Hospital Zurich, University of Zurich, 8006 Zurich, Switzerland; 4Experimental Center, Shandong University of Traditional Chinese Medicine, Jinan 250355, China; 5Shandong Co-Innovation Center of Classic TCM Formula, Shandong University of Traditional Chinese Medicine, Jinan 250355, China

**Keywords:** drug–drug interaction, organic cation transporter, pharmacokinetics, SLC, tyrosine kinase, TKI

## Abstract

Tyrosine kinase inhibitors (TKIs) decisively contributed in revolutionizing the therapeutic approach to cancer, offering non-invasive, tolerable therapies for a better quality of life. Nonetheless, degree and duration of the response to TKI therapy vary depending on cancer molecular features, the ability of developing resistance to the drug, on pharmacokinetic alterations caused by germline variants and unwanted drug–drug interactions at the level of membrane transporters and metabolizing enzymes. A great deal of approved TKIs are inhibitors of the organic cation transporters (OCTs). A handful are also substrates of them. These transporters are polyspecific and highly expressed in normal epithelia, particularly the intestine, liver and kidney, and are, hence, arguably relevant sites of TKI interactions with other OCT substrates. Moreover, OCTs are often repressed in cancer cells and might contribute to the resistance of cancer cells to TKIs. This article reviews the OCT interactions with approved and in-development TKIs reported in vitro and in vivo and critically discusses the potential clinical ramifications thereof.

## 1. Introduction

Standard chemotherapy represented the only pharmacological approach in the treatment of cancer until the beginning of the XXI century, when the shift to targeted therapies began with the development of the small-molecule tyrosine kinase inhibitor (TKI) imatinib, which was effective in the treatment of chronic myeloid leukemia and Philadelphia chromosome-positive acute lymphoblastic leukemia [1,2]. Since the approval of imatinib, more than 50 TKIs, heterogeneous in their structure and the targeted kinase, have been approved for clinical use, mainly for the treatment of solid tumors, and many others are in preclinical or clinical development [3]. The majority of TKIs have a common mechanism of inhibition, namely the competition with ATP at the nucleotide binding site of the kinase, which is located on the intracellular domain. However, other modes of inhibiting the signal transduction mediated by the tyrosine kinase include the competition of the TKI with the substrate for binding to the receptor, the competition with the phosphorylating entity or an allosteric negative cooperativity, in which TKI and substrate bind at two different non-overlapping sites, yet the binding of the TKI reduces the affinity of the receptor for its substrate. TKIs display substantial differences with respect to the spectrum of kinase inhibition, clinical application, pharmacokinetics and pharmacodynamics as well as the spectrum of possible drug–drug interactions (DDIs) [4]. Reversibility of the binding is one of the parameters used for categorizing targeted small-molecule TKIs. However, TKIs may further be grouped based on the target or the generation of belonging (e.g., first generation). In a recent review article, Roskoski described a thorough classification separating TKIs acknowledging the type (covalent or non-covalent) and site (e.g., front or back cleft) of the binding [5]. Most TKIs have been designed against membrane-bound receptor tyrosine kinases, which play an important role in a variety of cellular processes, including growth, motility, differentiation and metabolism, and are often aberrantly activated in cancer cells because of gain-of-function mutations, genomic amplification, chromosomal rearrangements and/or autocrine activation [6].

Growing evidence indicates that the expression and function of the polyspecific organic cation transporters belonging to the solute carrier (SLC) 22A family are critical to achieve therapeutic plasma and intracellular concentrations of some TKIs and are a hotspot of DDIs where TKIs are the causative drugs [4]. Organic cation transporters facilitate the movement across the plasma membrane of endogenous quaternary amines, such as tetraethyammonium (TEA), choline and L-carnitine, as well as several widely prescribed drugs, including platinum derivatives, metformin and dolutegravir [7,8,9]. To date, seven organic cation transporters have been characterized concerning their localization and substrate specificity. Conversely, the lack of obvious phenotypes associated with loss of, or gain of function of any of these transporters has hindered the comprehension of their physiological role. The human organic cation transporters 1, 2 and 3 (OCT1–3) are considered the canonical organic cation transporters and are encoded by the genes *SLC22A1*, *SLC22A2* and *SLC22A3*, clustering on chromosome 6 (6q25.3). OCT1-3 are Na^+^- and H^+^-independent transporters, whose activity is driven by the membrane potential and the electrochemical gradient of the substrate across the plasma membrane [10,11]. The human novel organic cation transporter 1 (OCTN1), encoded by the *SLC22A4* gene located on chromosome 5 (5q31.1), has been characterized as a carnitine/cation transporter driven by membrane potential [12]. However, there have been suggestions to rename it as ergothioneine transporter (ETT), to date, the substrate with the highest affinity for OCTN1 [13,14]. The gene *SLC22A5*, encoding for the human novel organic cation transporter 2 (OCTN2), clusters with the *SLC22A4* gene on 5q31.1. OCTN2 is a high-affinity Na^+^-dependent transporter of the quaternary amine L-carnitine and its precursor γ-butyrobetaine [15]. OCTN2 also transports several organic cations and drugs with low affinity and in a Na^+^-independent manner, including TEA, the antibiotic colistin, and oxaliplatin [11,16]. Among the organic cation transporters, OCTN2 is the only one with an indisputable physiological function. OCTN2 represents the main carnitine uptake system across epithelia and the loss of OCTN2 transport activity causes systemic carnitine deficiency (OMIM212149) [17]. Organic cation 6 (OCT6) is currently the preferred name of the transporter encoded by the *SLC22A16* gene on chromosome 6 (6q21-q22.1), initially named novel carnitine transporter 2 (CT2) [18]. Recently, the transporter encoded by the *SLC22A15* gene that was cloned alongside OCT6 was deorphanized but no functional name has yet been assigned to it. SLC22A15 transports zwitterions and cations, in most cases in a Na^+^-dependent manner [19].

As part of the Special Issue “Overcoming Biological Barriers: Importance of Membrane Transporters in Homeostasis, Disease, and Disease Treatment”, this review provides a detailed summary of the in vitro characterization of OCT-TKI interactions (Table 1) and describes the impact of the localization, expression level and function of OCTs on the pharmacokinetics, pharmacodynamics and toxicity of TKIs, as well as on the potential occurrence of DDIs, whereby TKIs inhibit the OCT-mediated transport of victim drugs (Table 2). Substrate specificity, molecular mechanism of transport and physiological role of organic cation transporters are marginally described and only as far as relevant to this review topic. For more details on the above-mentioned aspects, we refer to other review articles written by Koepsell [20,21], Tamai [22], Pochini et al. [23] and Samodelov et al. [11].

## 2. Organic Cation Transporter 1 (OCT1)

### 2.1. Expression in Normal and Tumor Tissues

The main site of expression of OCT1 in the human body is the sinusoidal membrane of hepatocytes. OCT1 is also expressed in the human intestine, but with uncertain subcellular localization [10,72,73,74]. Notably, in rodents, Oct1 is also highly expressed in the kidney, complicating the clinical translatability of the findings gathered from mouse studies [8]. Quantitative DNA methylation analysis showed that the promoter of the *SLC22A1* gene is often hypermethylated in hepatocellular carcinoma (HCC), as compared to the matched normal adjacent liver tissues of Caucasian patients, resulting in a significantly lower OCT1 protein level, assessed by immunohistochemistry, in HCC than in the normal liver [75]. Moreover, in the same study, it was found that the expression level of OCT1 inversely correlated with that of the proliferation marker Ki-67 [75]. The downregulation of OCT1 in liver cancer was confirmed in another study on 53 Caucasian patients diagnosed with primary HCC, in which it was found that the lower the expression level of OCT1, the worse the patient outcome, and that the degree of downregulation of OCT1 was associated with an advanced stage of the disease [76]. OCT1 protein level was found to be reduced also in an HCC Japanese cohort [77], suggesting that the downregulation of OCT1 is a hallmark of HCC, irrespective of ethnicity. In another study on Caucasian patients, OCT1 expression level was found to be significantly reduced in HCC but not in benign lesions, such as hepatocellular adenoma (HCA) and focal nodular hyperplasia (FNH) [78]. In the same study, the authors found that the expression level of OCT1 in the tumor relative to that in the surrounding normal liver tissue was associated with the pattern of accumulation and signal of the positron emission tomography (PET) tracer [^18^F]fluoromethylcholine, a substrate of OCT1 [78]. A clinical study on liver biopsies from 39 Caucasian patients with advanced HCC showed that OCT1 staining intensity was moderate to strong in 50% of the HCC samples analyzed without association with tumor stage. Notably, in the majority of the samples, staining was at the intracellular level. The authors hypothesized that nonsense mutations and aberrant splicing might generate truncated proteins with impaired trafficking to the plasma membrane [43]. There are conflicting data on the expression of OCT1 in leukemia cells. One research group could detect the mRNA of OCT1 in chronic myeloid leukemia (CML) human cell lines, in primary cultured cells from patients with CML and in blood cells from patients with CML [30,31]. Another research group reported that neither OCT1 mRNA nor the protein was detectable in CML cell lines and leukemic cells isolated from patients [65].

### 2.2. TKI Interactions In Vitro

Several TKIs have been shown to inhibit human OCT1-mediated transport in vitro, including imatinib, dasatinib, nilotinib, gefitinib, erlotinib, sunitinib, lapatinib, sorafenib [24], pazopanib [56] and saracatinib [40,41], indicating a relatively high drug–drug interaction (DDI) potential of TKIs at the level of OCT1. Based on the half-maximal inhibitory concentrations (IC_50_) of the uptake of the model substrate and the maximum unbound plasma concentration of the perpetrator TKI, only imatinib, nilotinib, gefitinib, erlotinib and pazopanib were considered clinically relevant inhibitors of OCT1 [24,56]. To notice that erlotinib IC_50_ of the uptake of the model substrate metformin was significantly lower for the M420del OCT1 in comparison with the wild-type OCT1 (IC_50_, 0.16 µM vs. 0.28 µM), suggesting that the loss of the methionine in position 420, which is associated with a lower metformin uptake rate, does not alter the binding to erlotinib [24]. Most TKIs inhibiting OCT1 transport activity in vitro were not transported by OCT1. It has been hypothesized that TKIs inhibit OCT1 by repressing a conserved phosphotyrosine switch. Experiments in HEK293 cells stably transfected with human OCT1 revealed a common indirect inhibitory mechanism by several TKIs that involves the inhibition of OCT1 phosphorylation by the Src-family kinase YES1 kinase, which phosphorylates OCT1 in positions Y240, Y361 and Y376. Because the incubation with TKIs did not affect OCT plasma membrane level, the authors hypothesized that the prevention of the phosphorylation might reduce binding site electronegativity and, thus, the interaction with cationic molecules [39]. However, structure–function and influx kinetic experiments are required to understand the role of these tyrosine residues in substrate binding to the transporter and the conformational changes thereof.

Several TKIs are potent inhibitors of OCT1, yet only a handful have been found to be substrates of OCT1. Experiments in HEK293 cells overexpressing human OCT1 demonstrated that pazopanib and saracatinib are weak substrates of OCT1 [41,51,56,65]. Conflicting results are available for sorafenib and imatinib. In one study, *Xenopus laevis* oocytes injected with human OCT1 cRNA were shown to incorporate 30% more sorafenib than water-injected oocytes [44]. Another study in HEK293 cells stably transfected with the murine or the human OCT1 showed that neither sorafenib nor the main metabolite sorafenib-N-oxide is transported by OCT1 [49]. With respect to imatinib, OCT1-transfected KCL22 cells incubated for 30 min with imatinib at an extracellular concentration of 5 µM showed a higher intracellular level of imatinib than that in mock-transfected cells [30]. In other studies, transport experiments performed in OCT1-expressing *Xenopus laevis* oocytes and in OCT1-HEK293 cells could not confirm imatinib as an OCT1 substrate [32,38,65].

### 2.3. TKI Interactions In Vivo

Oct1^−/−^ and wild-type mice treated with a single oral dose of 10 mg/kg sorafenib displayed comparable pharmacokinetic parameters of sorafenib and the main metabolite sorafenib-N-oxide [49]. A study in rats, singly dosed by gavage with metformin (100 mg/kg), sorafenib (100 mg/kg) or both, showed that sorafenib did not affect either the total exposure to metformin or its C_max_. However, sorafenib prolonged the time for metformin to reach the C_max_ (T_max_) [63]. Although individuals carrying the reduced-function variants of OCT1 displayed higher C_max_ and area under the plasma concentration–time curve (AUC) of metformin than those carrying the wild-type OCT1, with no changes in the T_max_ [79], it is possible that sorafenib inhibited OCT1-mediated intestinal absorption of metformin. With OCT1 being expressed on the brush-border membrane of enterocytes, sorafenib would inhibit the uptake of metformin from the intestinal lumen. In a scenario where OCT1 localizes on the basolateral membrane of enterocytes, OCT1 would mediate the exit step of the intestinal vectorial transport of metformin, namely the transport of metformin from the enterocyte cytosol to the portal venous system. In fact, although OCTs are often described as an uptake system, when the intracellular concentration of substrate is far greater than the extracellular one, OCTs can serve as an efflux system. Conversely, all pharmacokinetic parameters of sorafenib were comparable between the sorafenib and the sorafenib/metformin groups, confirming the in vitro findings, indicating that sorafenib is, at best, a weak substrate of OCT1 [63]. The lack of effect of metformin on sorafenib pharmacokinetics was confirmed in a phase I dose de-escalation study where patients were treated orally with 500 mg metformin twice a day and with increasing doses of sorafenib. Sorafenib steady-state plasma concentration was proportional to the dose from 200 up to 600 mg per day, with comparable risks of toxicity, suggesting that metformin is unlikely to alter the pharmacokinetics and the toxicologic profile of sorafenib [64].

Oct1/2^−/−^ mice injected with i.v. 50 mg/kg imatinib displayed plasma and hepatic level of drug comparable to that measured in wild-type animals. Similarly, plasma and hepatic levels of imatinib after oral gavage (50 mg/kg) were comparable between Oct1^−/−^ and wild-type mice, suggesting that OCT1 is not relevant in imatinib pharmacokinetics [65]. However, patients carrying an *SLC22A1* sub-haplotypic region encompassing three polymorphisms (rs3798168, rs628031 and IVS7+850C>T) were characterized by a significantly lower clearance and a higher trough concentration of imatinib than patients carrying other haplotype combinations [33]. In a study with 74 patients with c-KIT-positive gastrointestinal stromal tumors and treated with p.o. imatinib, the oral clearance of imatinib was comparable between patients expressing wild-type OCT1 and those carrying the reduced-function p.R61C (rs12208357) or p.G465R (rs34059508) variants [32]. In a small cohort of patients diagnosed with CML, the response to imatinib in R61C (rs12208357) heterozygote individuals was comparable to that achieved by patients carrying wild-type OCT1 [66]. In another study, OCT1 mRNA was detected in the blood of 70 imatinib-naïve patients diagnosed with chronic myeloid leukemia. Moreover, the patients with the highest pretreatment expression of OCT1 had a significantly better prognosis than those with low pretreatment expression of OCT1 [30]. Taken together, the data suggest that OCT1 is not relevant in imatinib pharmacokinetics, yet several clinical studies have found that OCT1 mRNA level is a good predictor of imatinib response [30,32,34]. A gene expression study in nine acute myeloid leukemia (AML) cell lines found a number of associations between the expression level of OCT1 and other transporters potentially involved in the uptake and efflux of imatinib. It is possible that OCT1 expression level is not causative of imatinib activity but rather a bystander that well correlates with the expression level of one or more transporters that transport imatinib in vitro (e.g., organic anion transporting polypeptide 1A2, P-glycoprotein) [32].

## 3. Organic Cation Transporter 2 (OCT2)

### 3.1. Expression in Normal and Tumor Tissues

OCT2 is predominantly expressed on the basolateral membrane of the renal tubule of humans and rodents [7,80,81,82,83,84,85]. Rigorous studies in rats indicated that Oct2 is mainly expressed on the basolateral membrane of proximal tubular cells of S1 and S2 segments, whereas the expression level fades in the S3 segment [86]. Other sites of OCT2 expression might be the human and murine dorsal root ganglia (DRG) [87], the rat choroid plexus [88], the inner- and outer-hair cells and stria vascularis of mouse cochlea [89]. The presence of OCT2 in the human ear can only be inferred from two pharmacogenetic studies that associated the rs316019 single-nucleotide polymorphism (SNP) (p.S270A), causing a reduced function in the transporter, with a lower risk of cisplatin-induced hearing loss (ototoxicity) [90,91]. A recent study showed expression of OCT2 in mouse pancreatic islet and in human INS-1 832-13 pancreatic β cells [92].

OCT2 expression is lost in both clear cell and papillary renal cell carcinoma (RCC), the two most frequent renal cell cancers [93]. In one retrospective study on 46 pairs of RCC tumors and adjacent normal tissues, it was found that OCT2 mRNA expression level was significantly decreased in RCC tumor tissues compared with that in adjacent normal controls [94]. A larger retrospective study on a tissue microarray containing 216 RCCs showed that OCT2 protein was practically absent in all RCC samples [85]. The repression of OCT2 in RCC seems to be of epigenetic nature. OCT2 promoter in RCC was characterized by hypermethylated CpG islands, the absence of H3K4 methylation and aberrant histone acetylation. Treatment of RCC cell lines with the hypomethylating agent decitabine and the histone deacetylase inhibitor vorinostat increased the expression of OCT2 and sensitized RCC cells to the OCT2 substrate oxaliplatin, both in vitro and in xenografts [94,95]. Another study demonstrated that OCT2 epigenetic activation by decitabine occurred in normoxic but not in hypoxic conditions [96]. Although OCT2 is not expressed in the intestine, in one study, the mRNA of OCT2 was detected in 11 out of 20 colon cancer samples [97].

### 3.2. TKI Interactions In Vitro

The OCT2-mediated uptake of model substrates, such as cisplatin, 4-Di-1-ASP(4-(4-(dimethylamino)styryl)-*N*-methylpyridinium (ASP^+^) and creatinine, was inhibited by clinically relevant extracellular concentrations (low micromolar range) of several FDA-approved TKIs, with dasatinib being one of the strongest inhibitors (IC_50_, 15.9 nM) [25,27,40,41,52]. Co-incubation with imatinib at an extracellular concentration of 10 µM significantly reduced the accumulation and cytotoxicity of the OCT2 substrate cisplatin used at an extracellular concentration of 50 µM [25]. Notably, most TKIs inhibiting OCT2 transport activity were not substrates of OCT2 [25,27,40,41]. Very much like for OCT1, the inhibition of OCT2 by TKIs is considered the result of the repression of YES1-mediated phosphorylation in positions Y241, Y362 and Y377 [27]. Moreover, the inhibitory effect of dasatinib on OCT2 transport activity was enhanced when cells were pre-incubated for 15 min with dasatinib [27], suggesting an atypical mechanism of inhibition that occurs only upon achieving inhibitory intracellular concentrations of dasatinib. In the same article, the authors found that OCTs have a proline-rich sequence that physically interacts with an Src Homology 3 (SH3) domain in YES1. Experiments of site-directed mutagenesis targeting the proline-rich SH3-binding domain of OCT2 showed reduced tyrosine phosphorylation and OCT2 transport activity [27]. Saracatinib and erlotinib are two exceptions and have been shown to be transported by OCT2. HEK293 cells stably transfected with the human OCT2 showed, respectively, 300% and 50% higher uptake of saracatinib and erlotinib than in wild-type cells [41,53].

### 3.3. TKI Interactions In Vivo

Wild-type mice treated with dasatinib displayed a higher plasma level and a reduced renal clearance of the OCT model substrate tetraethylammonium (TEA) in comparison with the vehicle-treated animals. Such a difference was not observed in mice lacking Oct2. The effect of Oct2 expression on dasatinib pharmacokinetics was not reported [27]. Similarly, Yes1^−/−^ mice showed an increased plasmatic TEA level in comparison with wild-type mice. Ex vivo experiments in DRG cells isolated from mice injected with vehicle or dasatinib showed that the cells exposed to dasatinib were more resistant to oxaliplatin-induced toxicity than those only treated with oxaliplatin [27]. In a randomized phase II clinical trial, erlotinib added to cisplatin and radiotherapy failed to significantly increase response rates or progression-free survival of patients diagnosed with locally advanced squamous cell carcinoma of the head and neck. Conversely, the combination appeared to be significantly safer than the cisplatin-alone treatment, concerning nephrotoxicity and ototoxicity. In the cisplatin arm (n = 96), there were six events of grade 3 or 4 nephrotoxicity or ototoxicity, whereas none were recorded in the cisplatin + erlotinib arm (n = 99) [70]. Mice lacking Oct2 have been previously shown to carry a lower risk of cisplatin-induced nephrotoxicity and otoxicity [89,98], and patients carrying the rs316019 SNP (p.S270A) showed a lower risk of nephrotoxicity and ototoxicity upon cisplatin therapy [90,91,99]. Thus, it is possible that the co-administration of erlotinib reduced cisplatin accumulation in proximal tubular cells and cochlear hair cells, without reducing the response to cisplatin, as OCT2 is probably not expressed in head and neck carcinoma.

## 4. Organic Cation Transporter 3 (OCT3)

### 4.1. Expression in Normal and Tumor Tissues

OCT3 has been identified and characterized as a high-capacity, low-affinity transporter of monoamines, thus, initially named extraneuronal monoamine transporter (EMT) [100]. OCT3 shows a wider expression throughout the body than OCT1 and OCT2. OCT3 mRNA and/or protein has been detected on the basolateral membrane of hepatocytes, proximal tubular cells, trophoblasts, brush-border membrane of enterocytes, plasma membrane of cardiomyocytes, brain and skeletal muscle cells [74,82,101,102,103,104,105]. The evidence that OCT3 plays a role in mood disorders and stress-related behaviors in animals [106] and in the handling of amphetamine [107,108] has led to further mapping the expression pattern of OCT3 in various areas of rodents’ amygdala, the area of the brain that drives emotional responses. OCT3 protein was found in both glial and neuronal perikaryon [109]. Retrospective clinical studies reported that OCT3 expression level was reduced in HCC [76] and in cholangiocarcinoma [110]. Noteworthily, OCT3 expression level in HCC was inversely correlated with that of OCT1 [76]. A recent retrospective study in patients with glioblastoma multiforme found that OCT3 protein level was significantly lower in tumor tissue than in the tumor-adjacent normal brain tissue and that patients with low OCT3 expression had a worse prognosis and outcome [111]. The mRNA of OCT3 was measured in normal and cancerous tissues from 16 patients diagnosed with colon or rectal cancer. In colon cancer, OCT3 mRNA level was significantly higher than that of the normal mucosa. No changes were found in rectal cancer [112].

### 4.2. TKI Interactions In Vitro

Imatinib, erlotinib, nilotinib and saracatinib have been shown to inhibit the uptake of ASP^+^, metformin, TEA or oxaliplatin in HEK293 cells stably transfected with human OCT3 [24,29,40,41]. The intracellular level of saracatinib in OCT3-HEK293 cells was three-times higher compared to the wild-type cells, suggesting that saracatinib is a substrate of human OCT3 [41]. A recent study systematically assessed the inhibitory effect of 25 TKIs at an extracellular concentration of 10 µM on the uptake of [^3^H]MPP^+^ in HEK293 cells overexpressing OCT3. Most TKIs exerted a significant inhibition of MPP^+^. Those with the strongest inhibitory effect were further investigated as an OCT3 substrate via trans-stimulation assay [42]. A transport cycle includes (i) the binding of the substrate to the transporter in the outward conformation, (ii) the inward conformational change and substrate translocation, (iii) the release of the substrate and (iv) the re-orientation of the carrier to its outward-facing conformation. The latter is the rate-limiting step when the reaction is carried in zero-trans conditions (no accompanying substrate) [113,114,115] and can be enhanced by performing the transport experiment in infinite-trans conditions, in which a substrate is present at both sides of the membrane. OCT3-HEK293 cells were pre-loaded with MPP^+^ or the TKI, washed and then the uptake of [^3^H]MPP^+^ was assessed. Unlike intracellular MPP^+^, which stimulated the uptake of extracellular [^3^H]MPP^+^ (homo-exchange), intracellular TKIs did not alter the uptake of extracellular [^3^H]MPP^+^ (hetero-exchange). The authors concluded that TKIs are strong inhibitors but not substrates of OCT3 [42]. Because the tyrosine residues and the proline-rich sequence responsible for the inhibition of OCT1 and OCT2 function by TKIs is also conserved in OCT3, it is possible that the molecular mechanism underlying OCT3 inhibition by TKIs is also related to the inhibition of YES1 activity.

### 4.3. TKI Interactions In Vivo

The anthracycline doxorubicin is an anticancer chemotherapeutic agent associated with irreversible cardiac injury that predisposes patients to an increased risk for congestive heart failure [116]. It has been recently shown that doxorubicin is a substrate of OCT3 and that it is expressed in patient-derived-induced pluripotent stem cell (iPSC) cardiomyocytes. The relevance of OCT3 in doxorubicin intracardiac accumulation is underscored by the finding that the level of doxorubicin in the heart of mice lacking Oct3 was significantly lower than that in the wild-type animals. Moreover, when wild-type animals were co-injected with nilotinib, a strong inhibitor of OCT3 transport activity, the doxorubicin level in the heart was significantly lower than that in the mice treated with doxorubicin only. In fact, in a doxorubicin-induced acute cardiac injury model, it was found that the loss of ejection fraction (measured by echocardiogram) and the increase in serum cardiac troponin I were less pronounced in the animals co-treated with nilotinib. Notably, nilotinib co-incubation did not reduce the cytotoxic effect of doxorubicin in a panel of leukemia and breast cancer cell lines [29].

## 5. Novel Organic Cation Transporters

### 5.1. Expression in Normal and Tumor Tissues

OCTN1 is widely expressed throughout the human body, with relatively high expression on the brush-border membrane of enterocytes, proximal tubular cells, plasma membrane of neural stem cells, neurons and microglia cells [117]. The mRNA and protein of OCTN1 were also found in human air epithelia [118]. A screening of the mRNA expression level of solute carriers in the NCI-60 cancer cell line panel (60 cancer cell lines) showed that OCTN1 was highly expressed in renal carcinoma cell lines and weakly expressed in colorectal and melanoma cancer cell lines [119]. Not much is known about the expression of OCTN1 in tumor tissues. Recently, a retrospective immunohistochemical analysis on specimens from 20 Chinese patients diagnosed with esophageal squamous cell carcinoma (ESCC) showed that OCTN1 was not expressed in either the esophageal normal mucosa or in ESCC [120].

Human OCTN2 is highly expressed on the brush-border membrane of enterocytes and proximal tubular cells and on the plasma membrane of skeletal muscle cells, brain capillary endothelial cells and brain cells [11,121,122]. OCTN2 mRNA and protein were found to be markedly increased in both squamous cell carcinoma and the adenocarcinoma of the esophagus. Moreover, patients treated with oxaliplatin who had high OCTN2 levels in the tumor tissue had a reduced risk of recurrence and a significantly longer survival time than those with low expression of OCTN2 in tumor tissue [120]. Another recent study showed that OCTN2 expression level was higher in more aggressive glioblastomas, hence, a negative prognostic marker in these patients [122], likely because of a higher accumulation of L-carnitine and, thereby, a higher rate of fatty acid β oxidation [123]. In fact, OCTN2 is the main L-carnitine cellular uptake system at physiological concentrations. Loss-of-function mutations in OCTN2 cause systemic carnitine deficiency in mice and in humans (OMIM212149) [17,124,125,126,127,128,129]. OCTN2 was also found to be expressed in a number of breast cancer cell lines and significantly higher in estrogen receptor (ER)-positive than in ER-negative tumor tissue specimens [130]. In the same study, the authors demonstrated that the presence of an intronic enhancer containing an estrogen-responsive element confers estrogen sensitivity to OCTN2 transcription [130].

OCT6 was cloned by Endou’s group from human testis cDNA libraries and characterized as a carnitine transporter localized on the luminal membrane of the epididymal epithelium and within the Sertoli cells of the testis [18]. At the same time, Moscow’s group cloned OCT6 using sequences from three fetal liver IMAGE clones. mRNA of OCT6 was measured, not only in the testis and fetal liver but also in bone marrow and peripheral blood leukocytes [131]. The mRNA of OCT6 was found in a number of cancer cell lines of different origins and in clinical samples from patients diagnosed with acute myeloid leukemia (AML) or acute lymphoblastic leukemia (ALL) [131,132,133]. OCT6 protein was found in the parental lung adenocarcinoma cell line PC-14 and in the human small-cell lung cancer cell line SBC-3 [134]. SLC22A15 was cloned from the total RNA of the human brain and human kidney. Data on tissue distribution and expression level in cancer can be found only in the Human Protein Atlas database. SLC22A15 mRNA was found in most normal tissues, with the highest level in the bone marrow and the brain. However, the SLC22A15 mRNA level was low or non-detectable in most cancers and cancer cell lines. For the less characterized OCT6 and SLC22A15, no data are yet available on the interaction with TKIs.

### 5.2. TKI Interactions In Vitro

The uptake of ASP^+^ mediated by the human OCTN1 is inhibited by saracatinib with an IC_50_ of 72 nM. Further transport experiments coupled to HPLC-UV detection showed that saracatinib was also a substrate of the human OCTN1 and transported with an influx K_m_ of 42  ±  11 µM and a maximal capacity (V_max_) of 79.7  ±  5.2 nmol/mg protein, and that the transport was significantly augmented in the presence of an outward proton gradient [41]. It should be pointed out that saracatinib, initially in the oncology pipeline of Astrazeneca, has been included in a drug repositioning program to test its efficacy for the treatment of other diseases, including rheumatoid arthritis, where several TK-dependent pathways are aberrantly activated [135]. It has been shown that OCTN1 is expressed in synovial fibroblasts isolated from tissues of patients with rheumatoid arthritis and that the uptake of saracatinib in these cells was reduced by 80% upon OCTN1 silencing [41].

*Xenopus laevis* oocytes injected with human OCTN1 or OCTN2 cRNA have been shown to incorporate more imatinib than water-injected oocytes [32]. The depletion of the Son of Sevenless 1 (SOS1), a guanine nucleotide exchange factor for Ras protein, significantly increases the expression of OCTN1 and the sensitivity of chronic myeloid leukemia cells to imatinib [136]. OCTN2-mediated transport of L-carnitine at an extracellular concentration of 10 nM was reduced by 50% in the presence of sunitinib at an extracellular concentration of 100 µM. Conversely, dasatinib, sorafenib, gefitinib and imatinib had no significant effect on L-carnitine OCTN2-mediated uptake. HEK293 cells overexpressing the human OCTN2 exposed to sunitinib at an extracellular concentration of 10 µM did not incorporate more sunitinib than wild-type HEK293 cells, suggesting that sunitinib is not a substrate of OCTN2 [137]. OCTN2 expression level in C2C12 myocytes was significantly reduced by treatment with lenvatinib [71]. *Xenopus laevis* oocytes injected with human OCTN2 cRNA have been shown to incorporate significantly more imatinib than water-injected oocytes [32].

### 5.3. TKI Interactions In Vivo

In a study on 54 Caucasian patients with unresectable/metastatic gastrointestinal stromal tumors (GISTs) in treatment with imatinib 400 mg daily as first-line therapy, it was found that the time to progression was significantly longer for patients carrying the OCTN1 rs1050152 variant (p.L503F) or the OCTN2 rs2631367 (g.-207G>C) and rs2631372 polymorphisms [67]. The OCTN1 p.Leu503Phe mutant has been shown to transport the substrate metformin more efficiently [138]. The g.-207G>C substitution was associated with a higher transcriptional activity of the promoter of the *SLC22A5* gene (encoding OCTN2) [139]. Conversely, patients diagnosed with chronic myeloid leukemia (CML) carrying the OCTN1 rs1050152 variant (p.L503F) had a lower probability of achieving a stable major molecular response to imatinib than the patients expressing wild-type OCTN1 [68]. Another pharmacogenetic study on patients diagnosed with CML found that patients carrying the OCTN1 rs460089 allele (g.813G>C) had a significantly higher probability of achieving a stable major molecular response to imatinib than the patients expressing wild-type OCTN1 [69]. Rats treated once a day for 14 days with lenvatinib administered by gavage showed a significant reduction in L-carnitine content in the skeletal muscle in comparison with the vehicle-treated animals, whereas plasma and urine levels were comparable between the treatment groups. Treatment with lenvatinib reduced the protein expression of OCTN2 in the skeletal muscle and the intestine of the rats but not in the kidneys [71].

## 6. Conclusions

The data gathered in this review article, mostly on OCT1 and OCT2, indicate that the majority of TKIs are poor substrates, if transported at all, by OCTs; hence, these transporters arguably play a marginal role in the pharmacokinetics and pharmacodynamics of TKIs. Conversely, several TKIs, irrespective of the targeted kinase and the clinical application, are potent inhibitors of OCT transport activity and, hence, clinically relevant perpetrators of DDIs, altering the pharmacokinetics and pharmacodynamics of other OCT substrates. While DDIs usually have a negative impact on the activity and tolerability of the victim drug, rationally designed DDIs might be beneficial for the treatment outcome. Whether DDIs with a TKI should be avoided or facilitated depends on the target organs of the victim drug. For instance, the inhibition of OCT1 is likely to reduce the anti-diabetic effect of metformin, whose accumulation in the liver, its site of action, is known to be primarily mediated by OCT1 [140]. Conversely, the inhibition of OCT2 might protect patients from nephrotoxicity and ototoxicity induced by platinum derivatives and the inhibition of OCT3 from doxorubicin-induced cardiotoxicity without reducing their antitumor properties, as both OCT1 and OCT3 are poorly expressed in cancer cells [29,52]. It was also revealed in this review article that the reduced expression level of OCT1 and OCT2 in hepatic and renal cancer, respectively, has been extensively associated with the pharmacological ramifications, namely drug resistance, but rather neglected with respect to the biological meaning of such a repression, yet this is an interesting aspect to elucidate, especially in light of the emerging role of OCTs in cell energetics [141,142,143].

## Figures and Tables

**Table 1 ijms-24-02101-t001:** OCT–TKI interaction in vitro.

Compound		OCT1	OCT2	OCT3	OCTN1	OCTN2
**Src-family**						
Imatinib	Inhibitor	IC_50_ = 1.47 µM ^1^ [24] IC_50_ (rat) = 4.1 µM ^5^ [25] Yes ^1^ [26]	Yes ^4^ [27] IC_50_ = 5.81 µM ^1^ [24] IC_50_ = 6.70 µM ^5^ [25] IC_50_ (rat) = 1.3 µM ^5^ [25] Yes ^4^ [28] Yes ^1^ [26]	IC_50_ = 4.36 µM ^1^ [24] IC_50_ (mouse) = 23.50 µM ^4^ [29] Yes ^1^ [26]	Not reported	Not reported
	Substrate	Yes [26,30,31,32,33,34,35,36,37] No [38]	No [26,32,38]	No [26,32,38]	No [32]	Yes [26,32]
Dasatinib	Inhibitor	IC_50_ = 1.07 µM ^1^ [24] IC_50_ = 0.20 µM ^1^ [39] IC_50_ = 0.56 µM ^4^ [39] Yes ^1^ [26]	IC_50_ = 15.9 nM ^3^ [27] Yes ^4^ [27] IC_50_ = 2.11 µM ^1^ [24] Yes ^4^ [28,39] Yes ^1^ [26]	IC_50_ = 4.50 µM ^1^ [24] Yes ^4^ [39] IC_50_ (mouse) = 17.30 µM ^4^ [29] Yes ^1^ [26]	Not reported	Not reported
	Substrate	Yes [37] No [26]	No [26]	No [26]		
Saracatinib *	Inhibitor	IC_50_ = 27.10 µM ^2^ [40] IC_50_ = 57.00 µM ^2^ [41]	IC_50_ = 0.46 µM ^2^ [40] IC_50_ = 0.90 µM ^2^ [41]	IC_50_ = 10.30 µM ^2^ [40] IC_50_ = 50.00 µM ^2^ [41]	IC_50_ = 72.00 nM ^2^ [41]	Not reported
	Substrate	Yes [41]	Yes [41]	Yes [41]	K_m_ = 42.00 µM [41]	
Nilotinib	Inhibitor	IC_50_ = 2.92 µM ^1^ [24] Yes ^1^ [26]	Yes ^4^ [27] IC_50_ > 30 µM ^1^ [24] Yes ^4^ [28]	IC_50_ = 0.35 µM ^1^ [24] IC_50_ (mouse) = 0.88 µM ^4^ [29] Yes ^1^ [26]	Not reported	Not reported
	Substrate	No [26,34]	No [26]	No [26]		
Ponatinib	Inhibitor	Not reported	Yes ^4^ [27,28]	Yes ^2^ [42]	Not reported	Not reported
	Substrate	No [26]	Not reported	Not reported		
Vemurafenib *	Inhibitor	Not reported	No ^4^ [28]	No ^2^ [42] IC_50_ (mouse) > 100 µM ^4^ [29]	Not reported	Not reported
	Substrate		Not reported	Not reported		
Bosutinib	Inhibitor	Yes ^4^ [39]	Yes ^4^ [27,28,39]	No ^4^ [39] Yes (mouse) ^4^ [29]	Not reported	Not reported
	Substrate	Not reported	Not reported	Not reported		
Sorafenib	Inhibitor	IC_50_ > 30 µM ^1^ [24]	No ^4^ [28] IC_50_ > 30 µM ^1^ [24]	IC_50_ = 20.10 µM ^1^ [24] Yes (mouse) ^4^ [29] Yes ^1^ [26]	Not reported	Not reported
	Substrate	Yes [43,44,45,46,47] Km = 3.80 µM [48] No [26,49,50]	Not reported	Not reported	No [26,50]	No [26,50]
Dabrafenib *	Inhibitor	Not reported	No ^4^ [28]	Not reported	Not reported	Not reported
	Substrate		Not reported			
Trametinib *	Inhibitor	Not reported	No ^4^ [28]	Not reported	Not reported	Not reported
	Substrate		Not reported			
**Epidermal growth factor receptor (EGFR)**						
Gefitinib	Inhibitor	IC_50_ = 1.07 µM ^1^ [24] Yes ^6^ [51] Yes ^1^ [26]	Yes ^4^ [27,28] IC_50_ = 24.40 µM ^1^ [24] Ye s^6^ [51] Yes ^1^ [26]	IC_50_ = 5.47 µM ^1^ [24] Yes (mouse) ^4^ [29] Yes ^1^ [26]	Not reported	Not reported
	Substrate	No [51] No [26]	Yes [51] No [26]	Not reported		
Erlotinib	Inhibitor	IC_50_ = 0.36 µM ^1^ [24] Yes ^1^ [26]	Yes ^4^ [27,28] IC_50_ = 5.24 µM ^1^ [24] Yes ^2+4^ [52] Yes ^1^ [26]	IC_50_ = 4.21 µM ^1^ [24] Yes (mouse) ^4^ [29] Yes ^1^ [26]	Not reported	Not reported
	Substrate	No [26]	Yes [26,53]	Not reported		
Afatinib	Inhibitor	Not reported	No ^4^ [28]	Yes ^2^ [42]	Not reported	Not reported
	Substrate		Not reported	Not reported		
Dacomitinib	Inhibitor	Yes [54]	Not reported	Yes ^2^ [42]	Not reported	Not reported
	Substrate			Not reported		
Lapatinib	Inhibitor	IC_50_ > 30 µM ^1^ [24] No ^4^ [39]	IC_50_ > 30 µM ^1^ [24] No ^4^ [28,39]	No ^2^ [42] IC_50_ > 30 µM ^1^ [24] No ^4^ [39] No (mouse) ^4^ [29]	Not reported	Not reported
	Substrate	Not reported	Not reported	Not reported		
Osimertinib	Inhibitor	Not reported	Not reported	Yes ^2^ [42] Yes (mouse) ^4^ [29]	Not reported	Not reported
	Substrate			Not reported		
**Platelet-derived growth factor receptor (PDGFR)**						
Sunitinib	Inhibitor	IC_50_ = 0.33 µM ^1^ [24] Yes ^4^ [39] Yes ^1^ [26]	Yes ^4^ [27,28,39] IC_50_ = 1.73 µM ^1^ [24] Yes ^1^ [26]	IC_50_ = 5.22 µM ^1^ [24] Yes ^4^ [39] IC_50_ (mouse) = 8.43 µM ^4^ [29] Yes ^1^ [26]	Not reported	Not reported
	Substrate	No [26,50]	Not reported	Not reported	No [26,50]	No [26,50]
Masitinib *	Inhibitor	IC_50_ (mouse) = 14 µM ^2^ [55] IC_50_ = 24 µM^2^ [55]	IC_50_ (mouse) = 19 µM ^2^ [55] IC_50_ = 7 µM ^2^ [55]	IC_50_ = 14 µM ^2^ [55]	Not reported	Not reported
	Substrate	Km = 16.30 µM [55] Yes (mouse) [55]	Yes [55]	Yes [55]		
**Vascular epithelial growth factor receptor (VEGFR)**						
Pazopanib	Inhibitor	IC_50_ = 0.25 µM ^1^ [56]	Yes ^4^ [27] IC_50_ = 3.5 µM ^2^ [57]	Yes (mouse) ^4^ [29]	Not reported	Not reported
	Substrate	K_m_ = 3.47 µM [56]	Not reported	Not reported		
Cabozantinib	Inhibitor	Not reported	Yes ^4^ [27,28]	No ^2^ [42] IC_50_ (mouse) = 5.93 µM ^4^ [29]	Not reported	Not reported
	Substrate		Not reported	Not reported		
Nintedanib	Inhibitor	Not reported	Not reported	Yes ^2^ [42]	Not reported	Not reported
	Substrate			Not reported		
Regorafenib	Inhibitor	Not reported	No ^4^ [28]	Yes ^2^ [42] Yes (mouse) ^4^ [29]	Not reported	Not reported
	Substrate	No [46]	Not reported	Not reported		
Axitinib	Inhibitor	Not reported	Yes ^4^ [27,28]	Yes (mouse) ^4^ [29]	Not reported	Not reported
	Substrate		Not reported	Not reported		
Vandetanib	Inhibitor	IC_50_ = 1.35 µM ^4^ [39] IC_50_ = 1.25 µM ^1^ [39]	Yes ^4^ [27,28,39] Yes [54] Yes^1^ [26]	Yes ^4^ [39] Yes (mouse) ^4^ [29]	Not reported	Not reported
	Substrate	Not reported	Not reported	Not reported		
Nindetanib *	Inhibitor	Not reported	Not reported	Yes (mouse) ^4^ [29]	Not reported	Not reported
	Substrate			Not reported		
**Cyclin-dependent kinases 4 and 6 (CDK4/6)**						
Abemaciclib *	Inhibitor	Not reported	Yes [54]	Yes ^2^ [42]	Not reported	Not reported
	Substrate			Not reported		
Ribociclib *	Inhibitor	Not reported	Not reported	Yes ^2^ [42]	Not reported	Not reported
	Substrate			Not reported		
**Bruton tyrosine kinase (BTK)**						
Acalabrutinib	Inhibitor	Not reported	Not reported	IC_50_ = 1.84 µM ^2^ [42]	Not reported	Not reported
	Substrate			Not reported		
Ibrutinib	Inhibitor	IC_50_ = 0.89 µM ^4^ [39] IC_50_ = 0.74 µM ^1^ [39]	Yes ^4^ [39]	Yes ^2^ [42] No ^4^ [39] IC_50_ (mouse) = 2.37 µM ^4^ [29]	Not reported	Not reported
	Substrate	Not reported	Not reported	Not reported		
Zanubrutinib	Inhibitor	Not reported	Yes [54]	Not reported	Not reported	Not reported
	Substrate		Not reported			
**Anaplastic lymphoma kinase (ALK)**						
Alectinib	Inhibitor	Not reported	Not reported	No ^2^ [42] No (mouse) ^4^ [29]	Not reported	Not reported
	Substrate			Not reported		
Brigatinib	Inhibitor	Yes [54]	Not reported	IC_50_ = 41.00 nM ^2^ [42]	Not reported	Not reported
	Substrate			Not reported		
Ceritinib	Inhibitor	Not reported	Yes [54]	IC_50_ = 28.20 nM ^2^ [42]	Not reported	Not reported
	Substrate			Not reported		
Crizotinib	Inhibitor	Not reported	Yes ^4^ [27,28] IC_50_ = 1.58 µM ^5^ [58] IC_50_ = 16.20 µM ^6^ [58]	IC_50_ = 0.11 µM ^2^ [42]	Not reported	Not reported
	Substrate		K_m_ = 1.16 µM [59]	Not reported		
Entrectinib	Inhibitor	Not reported	Not reported	Yes ^2^ [42]	Not reported	Not reported
	Substrate			Not reported		
Lorlatinib	Inhibitor	Not reported	Not reported	Yes ^2^ [42]	Not reported	Not reported
	Substrate			Not reported		
**Janus kinase (JAK)**						
Itacitinib *	Inhibitor	Not reported	IC_50_ = 5.8 µM ^5^ [60]	No ^2^ [42] IC_50_ = 80 µM ^5^ [60]	Not reported	Not reported
	Substrate		Not reported	Not reported		
Pacritinib *	Inhibitor	Not reported	Not reported	IC_50_ = 0.85 µM ^2^ [42]	Not reported	Not reported
	Substrate			Not reported		
Ruxolitinib	Inhibitor	Not reported	Yes ^4^ [27,28]	Yes ^2^ [42] No (mouse) ^4^ [29]	Not reported	Not reported
	Substrate		Not reported	Not reported		
Tofacitinib	Inhibitor	Not reported	No ^4^ [28] Yes [54]	Yes ^2^ [42] No (mouse) ^4^ [29]	Not reported	Not reported
	Substrate		Not reported	Not reported		
Fedratinib	Inhibitor	Not reported	Yes [54,61]	Not reported	Not reported	Not reported
	Substrate		Not reported			
Filgotinib *	Inhibitor	No ^4^ [62]	IC_50_ = 0.85 µM ^1^ [62]	Not reported	Not reported	Not reported
	Substrate	Not reported	Not reported			
Peficitinib *	Inhibitor	Yes [61]	Not reported	Not reported	Not reported	Not reported
	Substrate	Not reported				
**Human epidermal growth factor receptor (HER) family**						
Lenvatinib	Inhibitor	Not reported	Not reported	Yes ^2^ [42]	Not reported	Not reported
	Substrate			Not reported		
Neratinib	Inhibitor	Yes [54]	Not reported	Yes ^2^ [42]	Not reported	Not reported
	Substrate			Not reported		
**fms-like tyrosine kinase 3 (FLT3)**						
Gilteritinib	Inhibitor	IC_50_ = 0.02 µM ^4^ [39]	Yes ^4^ [39]	No ^4^ [39]	Not reported	Not reported
	Substrate	Not reported	Not reported	Not reported		

^1^ Inhibition of metformin uptake, ^2^ Inhibition of ASP^+^ uptake, ^3^ Inhibition of oxaliplatin uptake, ^4^ Inhibition of TEA uptake, ^5^ Inhibition of creatinine uptake, ^6^ Inhibition of MPP^+^ uptake. * Not approved [5].

**Table 2 ijms-24-02101-t002:** OCT-TKI interaction in vivo.

Compound	Type of Study	Dosage	Pharmacokinetics	Pharmacodynamics	Reference
Sorafenib	Oct1^−/−^ and wild-type mice	Sorafenib (p.o. 10 mg/kg)	C_max_ sorafenib: unchanged AUC sorafenib: unchanged	Not reported	[49]
	Wild-type rats	Metformin (p.o. 100 mg/kg) ± Sorafenib (p.o. 100 mg/kg)	C_max_ metformin: unchanged AUC metformin: unchanged T_max_ metformin: increased by sorafenib	Not reported	[63]
	Patients with advanced HCC	Sorafenib (p.o. 800 mg, 600 mg, 400 mg, 200 mg) + Metformin (p.o. 500 mg bid)	C_max_ sorafenib: unchanged	Risk of sorafenib-toxicity: positive correlation with sorafenib dose	[64]
Imatinib	Oct1^−/−^ and wild-type mice	Imatinib (p.o. 50 mg/kg)	C_max_: unchanged AUC: unchanged Hepatic accumulation: unchanged	Not reported	[65]
	Oct1/2^−/−^ and wild-type mice	Imatinib (i.v. 50 mg/kg)	C_max_: unchanged Hepatic accumulation: unchanged	Not reported	[65]
	Patients with CML	Imatinib (p.o. 300–600 mg)	Lower clearance and a higher trough concentration in patients with *SLC22A1* rs3798168, rs628031 and IVS7+850C>T haplotype	Not reported	[33]
	Patients with c-KIT^+^ GIST	Imatinib (p.o. 100–1000 mg)	Apparent oral clearance: no association with *SLC22A1* genotype.	Not reported	[32]
	Patients with CML	Not reported	Not reported	Therapeutic response: no association with *SLC22A1* genotype.	[66]
	Patients with CML	Not reported	Not reported	PFS and OS: positive association with OCT1 mRNA level.	[30]
	Patients with GIST	Imatinib (p.o. 400 mg)	Not reported	TTP: Association with *SLC22A4* AND *SLC22A5* genotype.	[67]
	Patients with CML	Imatinib (p.o. 400 mg or 800 mg)	Not reported	Therapeutic response: Association with *SLC22A1* and *SLC22A4* genotype.	[68]
	Patients with CML	Not reported	Not reported	Therapeutic response: Positive association with *SLC22A4* genotype and negative association with *SLC22A5* genotype	[69]
Dasatinib	Oct1/2^−/−^ and wild-type mice	TEA (i.v. 0.2 mg/kg) ± dasatinib (p.o. 15 mg/kg) Oxaliplatin (i.v. 10–40 mg/kg) ± dasatinib (p.o. 15 mg/kg)	Dasatinib increased plasma concentration and decreased excretion of TEA in wild-type but not in Oct1/2^−/−^ mice. Dasatinib did not alter the systemic disposition of oxaliplatin neither in wild-type or Oct1/2^−/−^ mice.	Risk of oxaliplatin-induced acute sensory neuropathy was reduced by dasatinib.	[27]
Erlotinib	Patients with locally advanced SCCHN	Cisplatin (i.v. 100 mg/m^2^ ± Erlotinib (p.o. 150 mg daily)	Not reported	PFS: unchanged Risk of cisplatin-induced nephrotoxicity and ototoxicity was lowered by erlotinib	[70]
Nilotinib	Wild-type mice	Doxorubicin (i.p. 3 or 5 mg/kg) ± Nilotinib (p.o. 15 mg/kg)	C_max_ doxorubicin: unchanged AUC doxorubicin: unchanged	Risk of doxorubicin-induced cardiotoxicity was reduced by nilotinib	[29]
Lenvatinib	Wild-type rats	Lenvatinib (p.o. 0.2–2mg/kg)	Lower L-carnitine content in the skeletal muscle but not in the plasma and urine.	Not reported	[71]

AUC = area under the plasma concentration–time curve; C_max_ = maximal plasma concentration; CML = chronic myeloid leukemia; GIST = gastrointestinal stromal tumor; HCC = hepatocellular carcinoma; OS = overall survival; PFS = progression-free survival; SCCHN = squamous cell carcinoma of the head and neck; TEA = tetraethylammonium T_max_ = time to C_max_; TTP = time to progression.

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
