# Peer review of "The Role of Organic Cation Transporters in the Pharmacokinetics, Pharmacodynamics and Drug–Drug Interactions of Tyrosine Kinase Inhibitors"

_ijms, 2023, doi:10.3390/ijms24032101_

Round 1

Reviewer 1 Report

The manuscript by Xiu et al. summarized interactions between TKI and OCTs in vitro and in vivo. This is of high interest due to the importance of TKIs in cancer management. The authors collected some data and summarized them into sections that highlight the TKI-OCT interactions. However, there are several issues to consider, as follows.

1. The authors should include a section introducing TKI inhibitors (e.g., structure, classification, clinical status, etc.)

2. The title of this manuscript emphasizes the role of OCTs in PK and PD of TKIs. Thus, the main focus is how PK and PD of TKIs are changed when OCT function is altered. This role needs to be discussed in sections 2.3, 3.3, 4.3, and 5.3 (TKI interactions in vivo with OCTs). However, there were few PK and almost no PD data of TKIs in these sections. For example, section 4.3 mentioned only one study describing the effects of nilotinib on the PKs of doxorubicin. Therefore, the authors should collect more data and include relevant discussion in these sections.

3. The authors should separate these sections into two parts: TKI-OCT interaction in vivo (demonstrated by effects of TKIs on PK and PD of OCT substrates) and OCT effects on PK and PD of TKI [shown by effects of inhibiting OCT (using OCT inhibitor or OCT knock-out/ knock-down animal models) on PK and PD of TKI].

4. The authors should include a table summarizing TKI interaction in vivo.

Author Response

We would like to thank the reviewers for the overall positive feedback. We agreed with most of the suggestions. The point-to-point answers can be found in the attached document.

Best regards

Reviewer 2 Report

In this paper the authors present a comprehensive literature review on tyrosine kinase inhibitors (TKIs) and their interactions with organic cation transporters (OCTs).

The manuscript is well-structured and provides interesting up-to-date information on the subject. However, agreeing with the authors' conclusion that OCTs have a "marginal role in the pharmacokinetics and pharmacodynamics of TKIs", I think that more emphasis should be placed on possible OCTs-associated drug interactions between TKIs and other drugs and maybe a new section should be added in the manuscript.

Furthermore, there are some minor syntax/grammatical errors than need to be addressed:

- Line 150: Please change "has" to "have"

- Line 170: Comma after "Although" should be omitted

- Line 173: Please change "inhibits" to "inhibited"

- Line 150: Please change "patients with a wild type OCT1" to "patients expressing wild type OCT1"

- Line 202: Please define which types of blood cells

- Line 210-211: Please change "correlate" to "correlates"

- Line 249: Please change "inhibitor" to "inhibitors"

- Please rephrase the sentence in lines 255-258: "Moreover, it was shown...model substrate [68]."

- Line 263: Please change "and shown to" to "known to"

- Line 265: Please change "3fold" to "300%"

- Line 271: Please change "in the mice lacking" to "in mice lacking"

- Line 272: Please change "in comparison with the" to "in comparison with"

- Line 283: Please remove "which is associated with a reduced OCT2 function", as it was already mentioned earlier in the text

- Line 287: Please change "carcinoma of the head and neck" to "head and neck carcinoma" 

- Line 300: Please change "has led to map further the expression pattern of" to "has led to further mapping the expression pattern of"

- Line 305- 306: Please change "multiform" to "multiforme"

- Line 315: Please change "in HEK293 stably" to "in HEK293 cells stably"

- Line 316-317: Please change "was three times that in the wild type" to "was three times higher compared to the wild type"

- Line 328: Please change "side" to "sides"

-Line 428: Please change "unrectable" to "unresectable"

Author Response

(The authors gave the same response as above.)

Round 2

Reviewer 1 Report

The manuscript was appropriately revised and can be accepted as is.